# Topological LC-circuits based on microstrips and observation of electromagnetic modes with orbital angular momentum

Yuan Li[1], Yong Sun[1], Weiwei Zhu[1], Zhiwei Guo [1], Jun Jiang[1], Toshikaze Kariyado [2], Hong Chen[1] & Xiao Hu[2]

New structures with richer electromagnetic properties are in high demand for developing novel microwave and optic devices aimed at realizing fast light-based information transfer and information processing. Here we show theoretically that a topological photonic state exists in a hexagonal LC circuit with short-range textures in the inductance, which is induced by a band inversion between $p$- and $d$-like electromagnetic modes carrying orbital angular momentum, and realize this state experimentally in planar microstrip arrays. Measuring both amplitude and phase of the out-of-plane electric field accurately using microwave near-field techniques, we demonstrate directly that topological interfacial electromagnetic waves launched by a linearly polarized dipole source propagate in opposite directions according to the sign of the orbital angular momentum. The open planar structure adopted in the present approach leaves much room for including other elements useful for advanced information processing, such as electric/mechanical resonators, superconducting Josephson junctions and SQUIDs.

[1] MOE Key Laboratory of Advanced Micro-Structured Materials, School of Physics Science and Engineering, Tongji University, Shanghai 200092, China. [2] International Center for Materials Nanoarchitectonics (WPI-MANA), National Institute for Materials Science, Tsukuba 305-0044, Japan. Correspondence and requests for materials should be addressed to H.C. (email: hongchen@tongji.edu.cn) or to X.H. (email: HU.Xiao@nims.go.jp)

To harness at will the propagation of electromagnetic (EM) waves constitutes the primary goal of photonics, the modern science and technology of light, expected to enable novel applications ranging from imaging and sensing well below the EM wavelength to advanced information processing and transformation. So far, systems with spatially varying permittivity and/or permeability, or arrays of resonators were explored, and EM properties unavailable in conventional uniform media have been achieved, such as negative refractive index, superlensing, cloaking and slow light[1–6], etc.

Inspired by the flourishing topological physics emerging in condensed matter[7–13], robust EM propagation at the edge of photonic topological insulators immune to back-scattering from sharp corners and imperfections came into focus in the past decade. This is achieved by one-way edge EM modes in systems with broken time-reversal symmetry (TRS)[14–23], and by pairs of counterpropagating edge EM modes carrying opposite pseudospins in systems respecting TRS[24–35] (for a recent review, see ref. [36]). Topological photonic systems with TRS which avoid the need of application of external magnetic fields—albeit at the price of sacrificing partially absolute robustness—attract increasing interest since they are more compatible with semiconductor-based electronic and optical devices. In a two-dimensional (2D) topological photonic crystal with $C_{6v}$ symmetry proposed recently[27], $p$- and $d$-like EM modes of opposite parities with respect to spatial inversion are tuned to generate a frequency band gap, and the sign of the orbital angular momentum (OAM) plays the role of an emergent pseudospin degree of freedom (for OAM and spin angular momentum, a related physical quantity, of EM modes in various circumstances, see previous works[37–40]). While EM modes with pseudospin up and down are degenerate in bulk bands due to TRS, thus hard to manipulate, they are separated into two opposite directions in the topological interface EM propagation, which can be exploited for realizing novel EM functionality. However, up to now only field strengths along the interface between photonic crystals distinct in topology and transmission rates at output ports have been measured. Details of pseudospin states of the topological EM propagation remain unclear in topological photonic systems explored so far, which hampers their advanced applications (the valley degree of freedom and related OAM have been revealed in a photonic graphene by selectively exciting the two sublattices in terms of interfering probe beams[41]).

In this work, we present the direct experimental observation on pseudospin states of unidirectional interface modes in topological photonic metamaterials. Based on the insight obtained by analyzing a lumped element circuit model with honeycomb-type structure, we propose that topological EM propagation can be achieved experimentally in a planar microstrip array, a typical transmission line in the microwave frequency band[42] constructed as a sandwich structure of bottom metallic substrate, middle dielectric film and top patterned metallic strips, which is common in various electronic devices. When the metallic strips form a perfect honeycomb pattern, linear frequency-momentum dispersions appear in the normal frequency EM modes, very similar to the Dirac cones in the electronic energy-momentum dispersions seen in graphene. Introducing a $C_{6v}$-symmetric texture with alternating wide and narrow metallic strips opens a frequency band gap. In addition, a band inversion between $p$- and $d$-like EM modes arises when the inter-hexagon strips are wider than the intra-hexagon ones, yielding a topological EM state mimicking the quantum spin Hall effect (QSHE) in electronic systems. Taking advantage of the planar and open structure of this metamaterial, we measure distributions of both amplitude and phase of the out-of-plane electric field along the interface between two topologically distinct microstrip regimes using near-field

techniques. EM waves are launched from a linearly polarized source located close to the interface. We resolve the weights of $p$- and $d$-like EM components in the interface modes and clarify their dependence on the source frequency swept across the bulk frequency band gap. We further map out the circulating local Poynting vectors and reveal explicitly the pseudospin states locked to the propagating directions. The simple structure of the present topological microstrip device displaying local OAM in its EM modes enables easy fabrication and on-chip integration, which is advantageous for harnessing EM transport inside the metamaterial, and potentially the system can be exploited for building novel microwave antennas which emit EM waves carrying OAM. Furthermore, the open 2D structure adopted in the present approach leaves much room for including other elements, such as electric/mechanical resonators, superconducting Josephson tunneling junctions, and SQUIDs, which are useful for advanced information processing.

## Results

**Topological phase transition in lumped element circuit.** As a simplified model of our system (see Fig. 1a, b), we begin with a lumped element circuit shown schematically in Fig. 1c. On-node capacitors with a uniform capacitance $C$ establish shunts to a common ground plane. Link inductors with inductance $L_0$

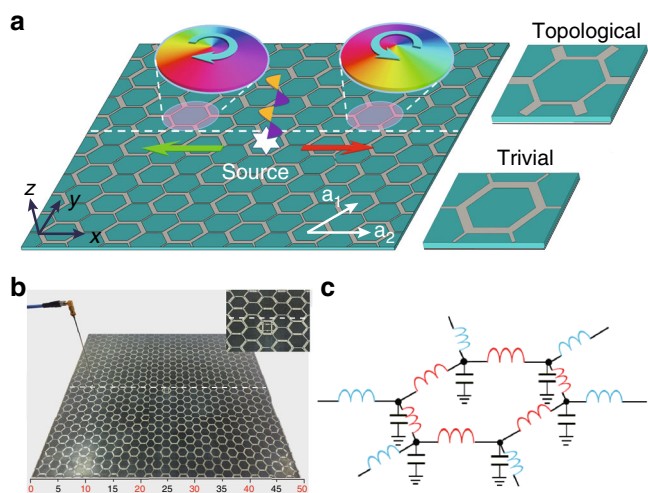

**Fig. 1** Design principle of microstrip-based topological LC-circuit. **a** Schematics of the honeycomb microstrip structure with enlarged views of the topologically nontrivial (upper) and trivial (lower) unit cells shown in the right panels. When excited by a linearly polarized source located in the interface with a frequency within the bulk frequency band gap, electromagnetic (EM) waves propagate rightward/leftward (red/green arrow) along the interface carrying up/down pseudospin, which is represented by the phase winding of the out-of-plane electric field $E_z$ accommodated in the hexagonal unit cells as indicated in the insets. **b** Photo of the experimental setup with a field probe placed right above the microstrip array, which is used to measure the distribution of the amplitude and phase of the out-of-plane electric field $E_z$, thereby resolving the pseudospin states and pseudospin-dominated unidirectional interface EM propagation. A lumped capacitor of $C = 5.6$ pF is loaded on the nodes. In the lower half of the system, the metallic strips of inter/intra hexagonal unit cell have widths of 1 and 2.6 mm, whereas in the upper half they are of 3.2 and 1.5 mm, respectively, and at the interface the width of metallic strips is taken as 2.6 mm. The length of all metallic strip segments is 10.9 mm and both lower and upper halves are composed of 14 × 8 hexagons. The whole microstrip system is fabricated on a F4B dielectric film with thickness of 1.6 mm and relative permittivity 2.2. **c** Schematic of the lumped element circuit of the hexagonal unit cells shown in the right panels of (**a**)

connect the nearest neighbor nodes within the honeycomb structure (drawn in red in Fig. 1c) and inductors with inductance $L_1$ (shown in blue in Fig. 1c) connect to the next hexagonal cell. Topological LC circuits were proposed and realized previously[34,35], in which cross-wirings with permutations were adopted to generate the nontrivial topology. In contrast, in the present approach, the nontrivial topology emerges purely from the symmetry of 2D honeycomb structure[27,43].

The voltage on a given node $i$ with respect to the common ground is described by (see Supplementary Note 1 for details)

$$d^2 V_i / dt^2 = \frac{-1}{C} \sum_{j=1}^{3} \frac{1}{L_{ij}} \left( V_i - V_j \right) \tag{1}$$

Taking the hexagonal unit cell shown in Fig. 1c, the normal frequency modes are governed by the following secular equation:

$$\left( 2 + \tau - \frac{\omega^2}{\omega_0^2} \right) \mathbf{V}_0 = Q \mathbf{V}_0 \tag{2}$$

$$Q = \begin{pmatrix} 0 & Q_k \\ Q_k^\dagger & 0 \end{pmatrix}, Q_k = \begin{pmatrix} \tau X Y^* & 1 & 1 \\ 1 & 1 & \tau X^* \\ 1 & \tau Y & 1 \end{pmatrix} \tag{3}$$

with $\mathbf{V} = \mathbf{V}_0 \exp(i\mathbf{k} \cdot \mathbf{r} - i\omega t) \equiv [V_1 V_2 V_3 V_4 V_5 V_6]^t \exp(i\mathbf{k} \cdot \mathbf{r} - i\omega t)$ for the voltages at the six nodes (see Supplementary Figure 1 for the numbering of nodes), where $X = \exp(i\mathbf{k} \cdot \mathbf{a}_1)$, $Y = \exp(i\mathbf{k} \cdot \mathbf{a}_2)$, $\omega_0^2 = 1/L_0 C$, $\tau = L_0/L_1$, and the asterisk means complex conjugating.

Figure 2 displays the frequency band structures for three typical values of $\tau$. As shown in Fig. 2a for $\tau < 1$, there is a global frequency band gap around 1.515 GHz, and the EM modes exhibit double degeneracy both below and above the frequency band gap. When $\tau = 1$, the frequency band gap is closed at the $\Gamma$ point, center of the Brillouin zone (BZ), and two sets of linear dispersions, known as photonic Dirac cones, appear with four-fold degeneracy at the $\Gamma$ point as displayed in Fig. 2b (accidental Dirac cones were achieved before in square lattices and were used to manipulate EM transport with zero refractive index[44]). For $\tau > 1$, a global frequency gap reopens as shown in Fig. 2c. While the frequency band structures in Fig. 2a, c look similar to each other, the EM modes in the two cases are different which can be characterized by the eigenvalue of the two-fold rotation operator $C_2$[45,46], or equivalently the parity with respect to the 2D spatial inversion, at the high-symmetry points of BZ (see Supplementary Figure 2 for details). For $\tau < 1$ (Fig. 2a), the parities of the eigen EM modes below the frequency band gap are given by "$+ - -$" at both $\Gamma$ and M points. In contrast, for $\tau > 1$ (Fig. 2c), the parities are given by "$+ + +$" at the $\Gamma$ point while "$- + -$" at the M point. With the parities of EM modes different at the $\Gamma$ and M points, the case $\tau > 1$ features a nontrivial topology. Therefore, the lumped element circuit exhibits a topological phase transition with the pristine honeycomb structure as the transition point.

The details of the phase windings at the $\Gamma$ point as displayed in Fig. 2d–g reveal that the EM eigenmodes can be designated as $p_\pm$ and $d_\pm$ orbitals. A $k \cdot p$ Hamiltonian can be formulated around the $\Gamma$ point based on these four orbitals[27], where the $4 \times 4$ matrix is block diagonalized into two $2 \times 2$ matrices, associated with the EM modes with positive and negative OAM, respectively (see Supplementary Note 2 for details). This $k \cdot p$ Hamiltonian takes the same form as the Bernevig–Hughes–Zhang model of QSHE proposed for HgTe quantum wells[12], where the two $2 \times 2$ blocks are associated with the electronic spin-up and -down states. Parallelizing these two Hamiltonians, it is clear that the sign of

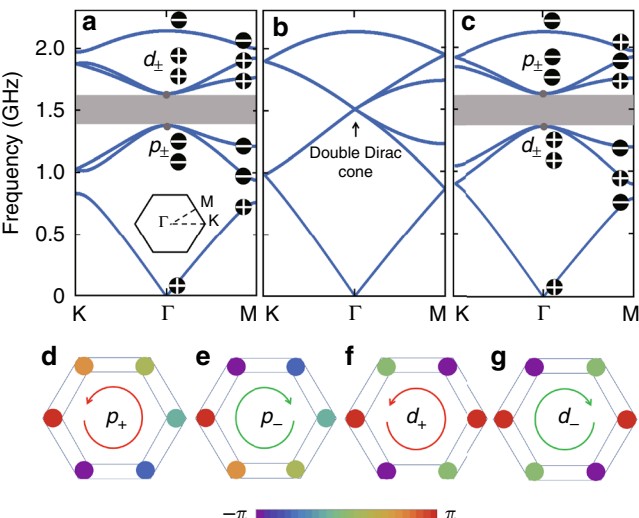

**Fig. 2** Band structures and topological phase transition in LC circuit. Frequency band structures calculated based on Equations (2) and (3) for **a** $\tau < 1$ with $L_0 = 3.60$ nH and $L_1 = 6.35$ nH, **b** $\tau = 1$ with $L_0 = L_1 = 4.22$ nH, and **c** $\tau > 1$ with $L_0 = 5.09$ nH and $L_1 = 3.13$ nH. The on-node capacitance is taken as $C = 7.27$ pF for all three cases. The distributed inductances and lumped capacitance are taken as tuning parameters, which reproduce the frequency band gap and the gap-center frequency of the experimental setup. The values of the distributed inductances are close to those evaluated from the experimental structures of microstrip arrays, and the on-node capacitance is slightly larger than the lumped one due to the distributed capacitances coming from the microstrip lines. The signs "+" and "−" inside the black dots denote the parities of the eigen EM modes with respect to the two-dimensional spatial inversion at the high-symmetry $\Gamma$ point and M point of the Brillouin zone. **d**–**g** Phase distributions of the out-of-plane electric field $E_z$ for the four eigenmodes at the $\Gamma$ point close to the frequency band gap at 1.515 GHz in (**a**, **c**)

OAM of the eigen EM mode in the present topological microstrip arrays plays the same role as the spin in spin–orbit coupled electronic systems, indicating that the sign of the OAM behaves as an emergent pseudospin degree of freedom[27–31,47–49]. In the case of Fig. 2c, band inversion between $p$ and $d$ orbitals takes place, which is thus topologically distinct from the case of Fig. 2a, in agreement with the conclusion derived from the analysis based on the eigenvalue of $C_2$.

While for simplicity and transparency, we describe the topological phase transition by a lumped element circuit with a two-valued inductance and uniform capacitance, the phenomenon is generic for circuits with textured capacitances and/or inductances. The system can also be reformulated in terms of propagating electric and magnetic fields with dielectric permittivity and magnetic permeability as relevant parameters. Therefore, the physics revealed here also applies to a broad class of planar networks of waveguides[50] including coaxial cables and striplines.

**Observing pseudospin and p–d orbital hybridization.** We then implement experimentally the topological photonic state revealed above by designing the planar microstrip arrays as shown in Fig. 1b. Because the distributed capacitances of the metallic strips with respect to the ground plate estimated following the standard procedure[42] are smaller than the lumped ones by one order of magnitude, the distributed capacitances can be incorporated to good approximation into the on-node capacitance, resulting in the lumped element circuit discussed above. The widths of the metallic strips in the trivial and topological designs are chosen in such a way that the two bulk systems give a common frequency

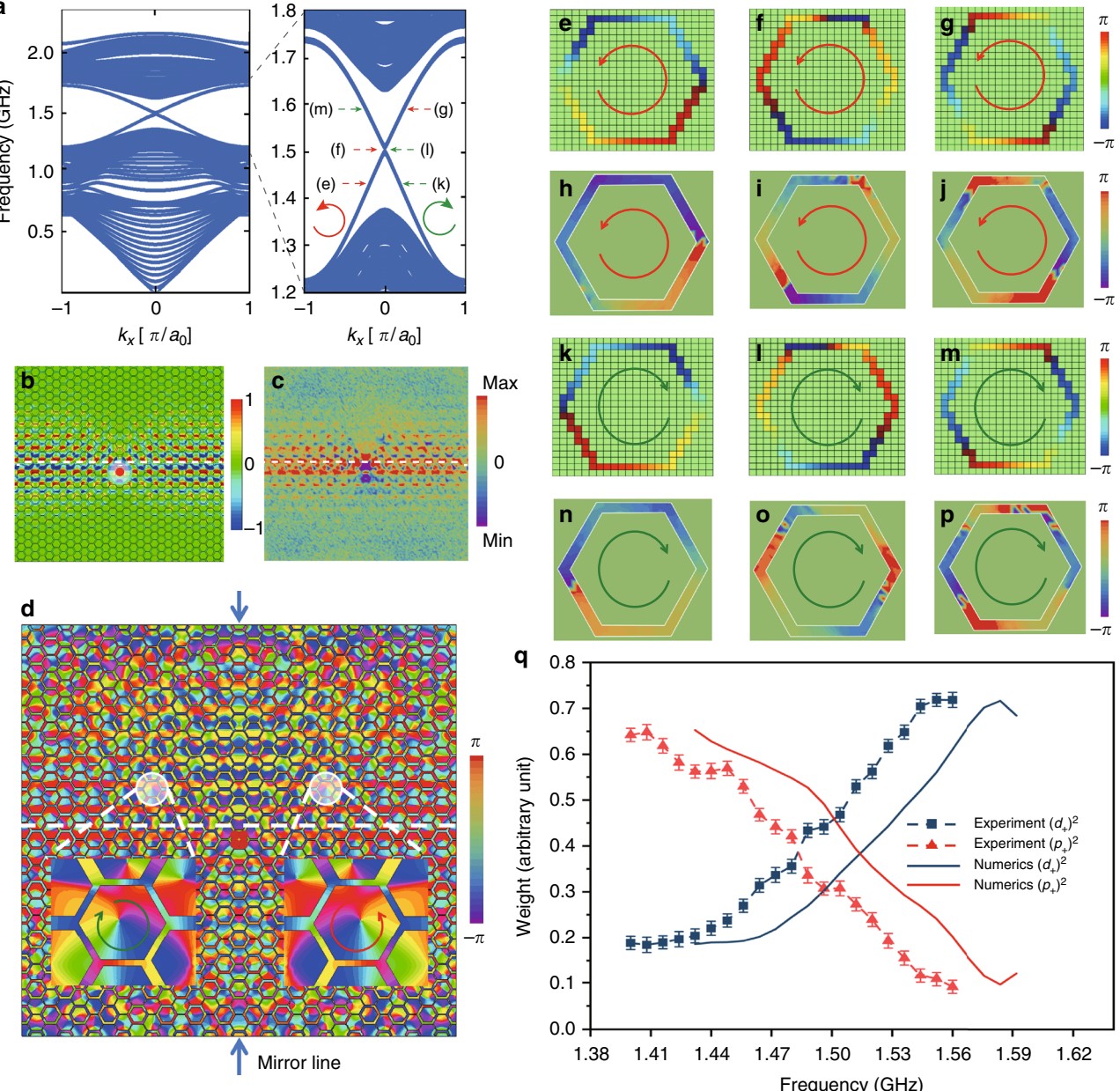

**Fig. 3** Resolving pseudospin and *p*–*d* orbital hybridization. **a** Calculated frequency band structure for the whole system in Fig. 1b with an interface between the topologically trivial and nontrivial regimes. A supercell is adopted including 8 hexagonal unit cells on both sides of the interface where the parameters for Fig. 2a, c are taken, respectively. The system is considered infinite along the direction of the interface. The right panel is a zoomed-in view of the frequency band diagram around the bulk band gap at 1.515 GHz. The red/green arrow indicates the dispersion of the rightward/leftward-propagating interface mode. **b, c** Distributions of the out-of-plane electric field $E_z$ obtained by the full-wave simulations (**b**) and experimental measurements (**c**) using a linearly polarized source located at the interface. The source frequency is set at $f = 1.47$ GHz for the full-wave simulations and $f = 1.44$ GHz for experimental measurements. **d** Phase distribution of the out-of-plane electric field $E_z$ under the same condition as (**b**) which is mirror symmetric with respect to the mirror line perpendicular to the interface indicated by the two dark-blue arrows. The source is located on the mirror line. The two insets show zoomed-in views of the phase distributions in the two typical hexagonal unit cells close to the interface, with the left/right one accommodating clockwise/counterclockwise phase winding. **e–g**, **h–j** Full-wave simulated and experimentally measured phase distributions in the right highlighted hexagon in (**d**) with up pseudospin at three frequencies indicated in the right panel in (**a**). **k–m**, **n–p** Same as those in (**e–g**) and (**h–j**), respectively except for the left highlighted hexagon in (**d**). **q** Frequency dependence of weights of *p* and *d* orbitals obtained by the full-wave simulations and experimental measurements for the right hexagonal unit cell in (**d**). The *p* and *d* orbitals take the same weight at the frequency where the two interface frequency dispersions cross each other in (**a**), with an apparent difference of 0.03 GHz between the simulated and experimental results. The error bars indicate statistical uncertainty (standard deviation) during three measurements

band gap, taking into account the common lumped capacitance. In order to reveal explicitly the topological EM properties, we put these two microstrip arrays side by side as displayed in Fig. 1a, b. As shown in Fig. 3a, obtained by numerical calculations based on

a supercell for the lumped element circuit, two frequency dispersions appear in the common bulk frequency gap due to the inclusion of the interface between the two half-spaces of distinct topology. It is interesting to note that these interface modes are

characterized mainly by two degrees of freedom, namely pseudospin and parity as resolved experimentally below.

In order to detect these topological interface EM modes experimentally, we launch an EM wave from a linearly polarized dipole source located in the interface with a frequency within the common bulk frequency gap (see inset of Fig. 1b). It is noticed that injecting an EM wave into the system without disturbing the bulk frequency band is a feature inherent to the bosonic property of photons which is not available for electrons. As displayed in Fig. 3b, c for the distributions of the out-of-plane electric field $E_z$ obtained by the full-wave simulations and experimental measurements (see Methods for details), respectively, the EM wave propagates only along the interface. Figure 3d shows the phase distribution of the out-of-plane electric field $E_z$ obtained by the full-wave simulations, which exhibits clockwise/counterclockwise phase winding in the half of the sample to the left/right of the source, as is clearly revealed by the mirror symmetry with respect to a mirror line perpendicular to the interface and passing through the source (indicated by the two dark-blue arrows in Fig. 3d). The two insets show the zoomed-in views of the phase distributions in the two typical hexagonal unit cells close to the interface, with the left/right one accommodating the clockwise/counterclockwise phase winding, which specifies the down/up pseudospin state of EM modes. This demonstrates a clear pseudospin-momentum locking in the topological interface EM propagation, mimicking the helical edge states in QSHE.

Now we investigate the variation of phase distribution in the topological interface EM modes when the source frequency is swept across the bulk frequency band gap. The interface EM modes intersecting the frequency bands below and above the band gap are composed from both $p$ and $d$ orbitals, which can be resolved by analyzing the phase winding noticing that for a $p/d$ orbital the phase winds $2\pi/4\pi$ over a hexagonal unit cell (see Supplementary Figure 3 and 4, and Supplementary Note 3 and 4 for details). As displayed in Fig. 3e–p obtained by the full-wave simulations and experimental measurements, at a frequency close to the lower band edge (Fig. 3e, h, k, n) the interface EM modes consist mainly of $p$ orbitals, and at a frequency close to the upper band edge (Fig. 3g, j, m, p) the interface EM modes are predominately $d$ orbitals, whereas $p$ and $d$ orbitals contribute equally at the center frequency of the band gap (Fig. 3f, i, l, o). Figure 3q displays the full frequency dependence of the weights of $p_+$ and $d_+$ orbitals evaluated in terms of the Fourier analysis on the phase distribution in the right zoomed-in hexagon in Fig. 3d (same results are obtained for the left hexagon and $p_-$ and $d_-$ orbitals as assured by the mirror symmetry), with a systematic frequency shift of 0.03 GHz between the experimental results and the ones obtained by the full-wave simulations due to the tolerance of the material and structural parameters in the fabrication. Because the $p$ and $d$ orbitals correspond to the dipolar and quadrupolar EM modes, respectively, using a linearly polarized dipole source we can generate and guide EM waves with the desired sign of OAM by choosing the propagation direction, and desired relative weight of dipolar and quadrupolar EM modes by choosing the working frequency within the topological band gap. These properties may be exploited to design topology-based microwave antennas and receivers.

The OAM accommodated in the hexagonal unit cell of the present microstrip array is intimately related to the local Poynting vector through the Faraday relation (see Supplementary Note 5 for details). For a harmonic mode with frequency $\omega$ the local Poynting vector is given by

$$\mathbf{S} = \mathrm{Re}[\mathbf{E} \times \mathbf{H}^*]/2 = \frac{|E_z|^2}{2\mu_0\omega}\left(\frac{\partial\varphi}{\partial x}\mathbf{x} + \frac{\partial\varphi}{\partial y}\mathbf{y}\right) \quad (4)$$

where $\mathbf{E} = E_z\mathbf{z} = |E_z|e^{i\varphi}\mathbf{z}$ (with $\mathbf{x}$, $\mathbf{y}$, and $\mathbf{z}$ being the unit vectors in the three spatial directions) and $\mathbf{H}$ are the out-of-plane electric

field and the in-plane magnetic field, respectively. For EM modes with fixed OAM such as $p_\pm$ and $d_\pm$ defined in the hexagonal unit cell, the local Poynting vectors circulate around the edges of the hexagon. It is obvious that $p_-$ and $d_-$ orbitals accommodate the Poynting vectors circulating clockwise whereas $p_+$ and $d_+$ orbitals accommodate those circulating counterclockwise, which correspond to the two pseudospin polarizations in the present system. One can evaluate explicitly the amount of angular momentum carried by the local Poynting vector given in Equation (4)

$$\mathbf{L} = \mathbf{r} \times \mathbf{S}/c^2 = \frac{|E_z|^2}{2\mu\omega c^2}\left(x\frac{\partial\varphi}{\partial y} - y\frac{\partial\varphi}{\partial x}\right)\mathbf{z}. \quad (5)$$

It can be shown (see Supplementary Note 5 for details) that for the EM mode $E_z = |E_z|\exp(i\varphi) = |E_z|\exp(il\theta)$ where $\theta$ is the azimuthal angle and $l = \pm1$ (for $p_\pm$) or $l = \pm2$ (for $d_\pm$) one has $\mathbf{L} = \frac{|E_z|^2}{2\mu\omega c^2}l\mathbf{z}$. Therefore, one photon of energy $\hbar\omega$ carries a quantized OAM $l\hbar$ along the normal of microstrip plane.

The local Poynting vectors measured experimentally at 1.44 GHz are displayed in Fig. 4a for the two typical hexagonal unit cells close to the interface, and those obtained by the full-wave simulations at 1.47 GHz are displayed in Fig. 4b over a wide region along the interface (shown schematically in Fig. 4c). Good agreement is achieved between experiments and simulations. On the left-/right-hand side of the source, the local Poynting vectors circulate clockwise/counterclockwise in the hexagonal unit cells in both topological and trivial regimes. In hexagons above the interface (i.e., in the topological regime), the density of the local Poynting vectors is larger at the bottom edge (closer to the

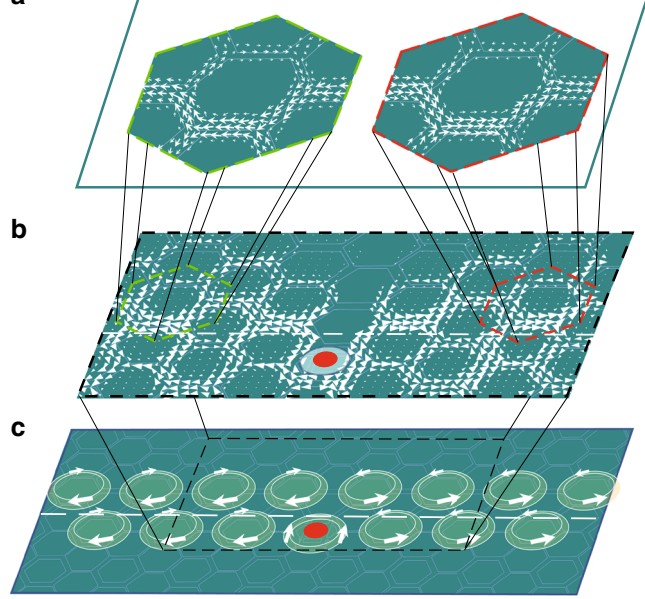

**Fig. 4** Distributions of local Poynting vectors in the interface EM modes. **a** Distribution of the local Poynting vectors obtained by experimental measurements of the amplitude and phase of the out-of-plane electric field $E_z$ (see Equation (4)) in the two hexagonal unit cells bounded by the dashed lines in (**b**) at 1.44 GHz. The size and direction of the arrows denote the amplitude and direction of the local Poynting vectors. **b** Distribution of the local Poynting vectors obtained by the full-wave simulations at 1.47 GHz in the region bounded by the dashed line in (**c**). **c** Schematic of the distribution of the local Poynting vector **S** over hexagonal unit cells in the topological interface modes stimulated by a linearly polarized source located in one of the unit cells (the red dots)

interface) than that on the top edge, which generates net energy flows in two directions along the interface. In hexagons below the interface (i.e., in the trivial regime), although the local Poynting vectors circulate in the same ways as those in the topological regime, the density of the local Poynting vectors is smaller at the top edge (closer to the interface) than that on the bottom edge, opposite to that in the topological regime, which therefore contributes the same net energy flows. This yields the winding phases, or equivalently the circulating local Poynting vectors, in hexagonal unit cells and the unidirectional energy flow along the interface. The distribution of the local Poynting vectors in Fig. 4 indicates explicitly that OAM of EM mode governs the topological interface EM propagation in the present system.

As seen above, the planar and open structure of the present system permits us to observe directly the pseudospin states, pseudospin-momentum locking, and furthermore the $p$–$d$ orbital hybridization in the interface EM modes, which constitute the essence of the topological state preserving TRS. So far, the relevance of pseudospin in topological interface propagation has been inferred based on comparisons between experimental observations and theoretical analyses, and there are few experimental studies on the relative weight of orbitals with opposite parities in topological interface propagations.

## Discussion

EM waves with OAM attract considerable current interest. It becomes clear that optical fields with OAM are ideal for many important applications such as communications, particle manipulation, and high-resolution imaging[51]. Even at microwave frequency, tunable OAM provides a new degree of freedom, which can be used for controlling on-chip propagation and can be exploited to develop novel high information density radar and wireless communication protocols[52,53]. The OAM explored in the present work is defined in unit cells and is oriented perpendicular to the propagating direction of the topological interface EM modes, different from OAM carried by light vortices in continuous media that is parallel to the propagating direction. To figure out a way to emit efficiently EM modes carrying OAM supported by the microstrip structure with $C_{6v}$ symmetry into free space is one of the most intriguing future problems. The topological EM properties achieved in the planar circuit can not only be exploited for microwave photonics[54] and plasmonics[55], but can also be extended up to the infrared frequency regime[56]. In the photonic wire laser working in the terahertz band[57,58], a patterned, double-sided metal waveguide is used for confining and directing the emission from a quantum-cascade laser, with the metal–semiconductor–metal structure essentially being the same as the microstrip array investigated in the present study. In terms of a honeycomb-patterned network with typically micrometer strip-widths one can achieve a topological quantum-cascade laser, where the emission and propagation of terahertz EM waves are governed by OAM. It is also worth noticing that the 2D structure of the present scheme makes the topological microwave-guiding compatible with various lithographically fabricated planar devices. Extension of the lumped element circuit discussed in the present work to a network including resonators of quantum features, such as quantum bits (qubit) based on SQUID structures[59], is of special interest.

## Methods

**Preparation of the sample and the full-wave simulations**. To prepare the perimeter of the whole sample, we load lumped resistors between the metallic strips and the common ground plane (i.e., bottom metallic substrate), which corresponds to a perfect matching boundary condition. The values of lumped resistors are selected according to the characteristic impedances $Z_0$ of microstrip lines, 115, 74, 97, and 66 Ω for microstrip lines with widths of 1.0, 2.6, 1.5, and 3.2 mm, respectively[42]. In order to numerically simulate the system, we perform three-dimensional full-wave finite-element simulations using Computer Simulation Technology Microwave Studio software based on a finite integration method in the time domain. The dielectric loss tangent (tan δ) of the substrate and the conductivity of the metallic microstrip lines are set to be 0.0079 and $5.8 \times 10^7$ S/m, respectively. The internal resistance in each lumped capacitor is taken as 1 Ω. An open boundary condition is applied for the whole sample including the terminal resistors. In order to demonstrate fully the unidirectional interface EM transport governed by pseudospin, we also prepare interfaces with sharp turns and stimulate the system by a source with signals overlapping exclusively with one of the two pseudospins of the interface modes (see Supplementary Figure 5 and Supplementary Note 6 for details).

**Experimental setup**. Signals generated from a vector network analyzer (Agilent PNA Network Analyzer N5222A) are transported into a port located in the sample, which works as the source for the system (see Supplementary Figure 6 and Supplementary Note 7 for details). A small homemade rod antenna of 2 mm length is employed to measure the out-of-plane electric field $E_z$ at a constant height of 2 mm from the microstrip lines. We make sure by the full-wave simulations that the field distribution thus measured is almost the same as that at the very surface of microstrip lines. The antenna is mounted to a 2D translational stage to scan the field distribution over the whole system with a step of 2 mm. A finer step of 1 mm is taken in order to measure accurately the field distribution in several typical hexagonal unit cells, which reveals the pseudospin structure. The measured data are then sent to the vector network analyzer. By analyzing the recorded field values, we obtain the distributions of both amplitude and phase of the out-of-plane electric field $E_z$, which are used for analysis of detailed phase windings, weights of $p$ and $d$ orbitals, and local Poynting vectors in the topological interface states.

**Code availability**. All the computer codes that support the findings of this study are available from the corresponding authors upon reasonable request.

## Data availability

The data that support the findings of this study are available from the corresponding authors upon reasonable request.

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

## Acknowledgements

H. Chen and Y. Sun are supported by the National Key Research Program of China (No. 2016YFA0301101), the National Natural Science Foundation of China (Grant Nos. 11234010, 61621001, and 11674247), the Shanghai Science and Technology Committee (Nos. 18JC1410900 and 18ZR1442900), and the Fundamental Research Funds for the Central Universities. X. Hu is supported by Grants-in-Aid for Scientific Research No.17H02913, Japan Society of Promotion of Science.

## Author contributions

Y.L. prepared the sample and conducted experimental measurements and the full-wave simulations. Y.S., W.Z., Z.G., and J.J. helped with experiments. T.K. joined discussions on the model and theoretical analyses. H.C. and X.H. conceived the idea, supervised the project, and wrote the manuscript. All authors fully contribute to the research.

## Additional information

**Competing interests:** The authors declare no competing interests.

