## [Peer Review File · Nature Communications]

Reviewers' comments:

Reviewer #1 (Remarks to the Author):

The authors present theoretical and experimental results which show a topological photonic state can be implemented in the microwave regime using a planar microstrip design. They claim that the directionality of the mode can be selected by the sign of the orbital angular momentum of light. For this, they adopted the idea of using hexagonal dielectric photonic crystal proposed by Wu and Hu (ref. 27). Additionally, they introduced a tuning parameter ($\tau = L1/L2$) to achieve topological transition from a topologically trivial state to a nontrivial state.

In my opinion, the main idea of using a microstrip design is of interest to researches in photonics and microwave engineering. The authors successfully showed the existence of topological photonic state in their structures. However, I think that the contribution of this paper is limited to experimental verification of the topological nature of honeycomb array in the microwave regime and the paper does not add useful insight on topological photonics or improved functionality such as active tunability. More importantly, it is not clear how the orbital angular momentum plays a role in the excitation of the chiral edge modes. Therefore, I do not think that the manuscript is appropriate for publications in Nature Communications.

1) The main drawback of the paper is that the orbital angular momentum (OAM) is not appropriately considered and only spin angular momentum (circular polarization) is considered for the light source. The authors should clarify what OAM they talk about and how the OAM can be quantified for their simulations and experiments (see K.Y. Bliokh, et al. Nature Photonics vol. 9, 796 (2015)). If they mean the extrinsic OAM which can be defined as an integral of $(\mathbf{r} \times \mathbf{k})$ or $(\mathbf{r} \times \mathbf{S})$ for the 2D space, more details should be included in the paper.

2) As shown in Fig. 4 and described in Ref 27, the selection of the clockwise/anti-clockwise modes is determined by the spin angular momentum (circular polarisation states). However, the authors state in the introduction that the sign of OAM plays the role of an emergent pseudospin degree of freedom, which can mislead and should be justified. In general, there is a close relation between the SAM and OAM (for example, R. Kerber et al., ACS Photonics vol. 4, 891 (2017)), and how the relation affects a light-matter interaction depends on the specific system considered.

3) The expression "fine tunable orbital angular momentum" in the title can mislead because the tuning parameter changes the topological state not the OAM. It also gives an impression that the system is actively tunable for example by optical or electrical stimuli, but in fact the tuning parameter is a fixed design parameter that determines which topological regimes a microstrip structure is.

4) It is interesting to see the direct measurement of topological phase transition, pseudo-spin dependent excitation of edge modes for the honeycomb structures but it is not surprising to see them because the mechanism and related symmetry discussions were already included in the paper by Wu and Hu (ref. 27).

Reviewer #2 (Remarks to the Author):

Referee report for manuscript titled:

"Generating electromagnetic modes with fine tunable orbital angular momentum by planar topological circuits"

By Yuan Li et al.

Photonic topological insulators have attracted much attention in the past decade. The unique transport properties of edge modes in topological systems allow for robust propagation, immune to scattering from defects and unconventional trajectories such as corners. This robust propagation is very attractive for applications in nano-photonics and photonic circuits, allowing novel designs that were previously thought of as impossible. True topological protection requires breaking of time reversal symmetry, usually done by applying an external magnetic field. However, the magnetic field constitutes a significant hurdle in implementing structures, especially in the optical domain. While photonic topological insulators without magnetic fields offer topological protection that is not pure, they are more viable as platforms for applications. Photonic topological insulators of both kinds have been experimentally demonstrated by several groups over the past years.

This manuscript presents an experimental demonstration of a crystalline topological insulator – in which the topological protection is based on crystalline symmetry, without breaking time reversal symmetry. The experiments are performed in the microwave regime using microstrip transmission line structure. There are two main novel claims of the paper: One, the implementation of a photonic topological insulator in a microstrip structure, which is an open two-dimensional structure allowing for lithographic manufacturing and thus integration to more complex systems. Two, the direct measurement of the intensity and phase of the propagating edge modes, emphasizing the role of OAM in the structure.

Given the increasing interest in photonic topological insulators, I believe the manuscript is of interest to the community. Specifically, structures that allow for future miniaturization and integration are of interest, and this manuscript provides one such example. The manuscript is clearly written and the results are presented in a convincing manner, both to experts in the field and a general audience.

Therefore my conclusion is that the manuscript should be published in Nature communications.

Minor points:

- In line 36 and 37, the authors mention applications for photonics in general. They mention IoT (Internet of Things. I had to search for the acronym – it was not obvious to me) and self-driving automobiles. These seem to me too broad applications, without any immediate relation to metamaterials or photonic circuits. The authors should elaborate.

- In Figs 3 and 4 the simulation and experimental results are shown in low resolution. While I could understand the results, I had trouble doing so. I assume the finished manuscript will have higher resolution figures so the reader can clearly observe the phase structure of the EM fields.

Reviewer #3 (Remarks to the Author):

The authors studied topological effects in planar circuits and realized different relative weights of p- and d-orbital-like EM modes by sweeping frequency. I have no formal objection to the results presented in the manuscript, but this research is a routine development in this area by considering:

- 1) The theoretical claim of topological band gap in C₆-symmetric systems had been presented in the same authors' previous PRL paper, and it has been adopted to many different photonic systems, e.g., refs. 28-30, and even phononic systems [e.g., Nature Physics 12, 1124 (2016) & Physical Review Letters 119, 255901 (2017) & Physical Review B 97, 020102 (2018), etc]. The planar circuits discussed in this manuscript is just another implementation. It is not timely and will not of interest to others in the community and the wider field.
- 2) The experimental results presented in Fig. 4 had been demonstrated in authors' previous work [ref. 30]. Both of them are microwave observation, and once again this results are expected.
- 3) The authors try to claim that the so-called tunable orbital angular momentum by sweeping frequency. However, I doubt the claim of "such phenomena can hardly be achieved in other photonic systems" in the Abstract. Based on the same theory, similar phenomena should be found in the C₆-symmetric all dielectric photonic crystals.
- 4) Why is the tunable orbital angular momentum important? There is no answer from the text.
- 5) Orbital angular momentum is important in the optical frequency, can this planar circuit be extended to optical frequency?
- 6) The paper is hard to read and bad to organize. For example:
 - i) in Line5 Page4, it is so confused to have optical concept of OAM in the microwave regime. Should clarify the connection between optics and microwave.
 - ii) why the author claimed 2D structure by described the structure consisting of three layers?
 - iii) there is an offset of 0.03G between numerical and experimental results in Fig. 3. But the explanation is "due to the tolerance of the material and structural parameters in the fabrication", it is too simple to convince reader.
 - iv) the sum of $(d+)^2 + (p+)^2$ is not equal to 1 [see Fig. 3q]?
 - v) the amplitude of E_z is missing in the calculation of the weight of p and d orbital states.

Based on the above reasons, I do not recommend this work for publication of Nature Communications.

Reviewer #1 (Remarks to the Author):

Comment: The authors present theoretical and experimental results which show a topological photonic state can be implemented in the microwave regime using a planar microstrip design. They claim that the directionality of the mode can be selected by the sign of the orbital angular momentum of light. For this, they adopted the idea of using hexagonal dielectric photonic crystal proposed by Wu and Hu (ref. 27). Additionally, they introduced a tuning parameter ($\tau = L1/L2$) to achieve topological transition from a topologically trivial state to a nontrivial state.

Reply: We thank this Reviewer for careful reading of our manuscript.

Comment: In my opinion, the main idea of using a microstrip design is of interest to researchers in photonics and microwave engineering. The authors successfully showed the existence of topological photonic state in their structures. However, I think that the contribution of this paper is limited to experimental verification of the topological nature of honeycomb array in the microwave regime and the paper does not add useful insight on topological photonics or improved functionality such as active tunability. More importantly, it is not clear how the orbital angular momentum plays a role in the excitation of the chiral edge modes. Therefore, I do not think that the manuscript is appropriate for publications in Nature Communications.

Reply: We appreciate this Reviewer for her/his positive assessment that our work based on microstrip is of interest to researchers in photonics and microwave engineering. Meanwhile s/he raises questions on new insight or improved functionality such as active tunability, and points out specifically that the role of orbital angular momentum in exciting chiral edge modes has not been explained in a clear way. All these are key points of our present work.

We respectively disagree with her/his comment that the contribution of this paper is limited to experimental verification of the topological nature of honeycomb array in the microwave regime. Because the design principle is based on $C6v$ symmetry, the present approach does share some common features with a few previous works. However, this does not degrade the novelty of the present work for the following reasons. Taking advantage of the planar and open structure of the microstrip, we have successfully detected the orbital angular momentum (OAM) of EM modes and clarified in a direct

experimental way its importance in governing the unidirectional topological interface propagation, which can hardly be achieved in any other electronic and bosonic systems.

As is well known, transmission line (TL) is a basic building block ubiquitous in almost all electromagnetic devices, and microstrip is a typical TL consisting of metallic substrate, middle dielectric film and top metallic strip. The present work demonstrates clearly that arranging the metallic strip into a honeycomb pattern merely infuses nontrivial Berry phase and topology into the EM modes. It is remarkable that one does not have to pay any additional price to get the valuable topological features in TL. The planar circuit structure investigated in the present work can be fabricated with conventional techniques, which can be pushed all the way down to nanoscale where advanced lithographical fabrications are already available. All these merits make the approach adaptive to various existing devices, and leave much room for exploration of new functionalities. It is also worth mentioning that the underlying physics is captured by simple LC circuitry, therefore the recipe is accessible to most engineers, and even by high school students. The results are not only useful directly for microwave and photonic engineering, but also suggestive for other circuitry systems including those used for controlling qubits by microwaves or surface mechanical waves. With potential advantages in miniaturization and integration, we believe the present work will trigger various applications towards realizing new functionality and thus has far reaching importance.

In what follows answer all comments one by one.

Comment 1) The main drawback of the paper is that the orbital angular momentum (OAM) is not appropriately considered and only spin angular momentum (circular polarization) is considered for the light source. The authors should clarify what OAM they talk about and how the OAM can be quantified for their simulations and experiments (see K.Y. Bliokh, et al. Nature Photonics vol. 9, 796 (2015)). If they mean the extrinsic OAM which can be defined as an integral of $(r \text{ cross } k)$ or $(r \text{ cross } S)$ for the 2D space, more details should be included in the paper.

Reply: We thank this Reviewer for raising this important comment, and for indicating explicitly the way to improve presentation. We wish to resolve her/his concern in what follows.

As in conventional microstrips, the EM mode under concern propagates in the lateral direction, with the electric field pointing in the normal direction and the magnetic field aligned laterally. For such a mode (defined as TM mode in our manuscript), one cannot define the conventional circular polarization. In this sense, as can be seen in Box 1 in Fig. a in Box 1, the spin angular momentum (SAM) discussed in the paper by K.Y. Bliokh et al. (Ref. 38 in the updated manuscript) is not directly relevant. The pseudospin in the present approach emerges from the periodic permittivity in two dimensions (2D) with C6v symmetry, and governs the direction of 1D topological interface propagation.

In addition to results presented in the last version of our manuscript, such as those in Figs. 2 and 3, we provide detailed discussions on the eigen EM modes defined in hexagonal unit cells and the corresponding local Poynting vectors, which responds directly to the comment from this Reviewer and enhances our statements. Considering the harmonic EM mode with given frequency ω , we arrive at an eigenvalue problem as given by Eq. (2) in terms of the single z-component of electric field E_z . In accordance with the C6v symmetry of the honeycomb-based structure, the eigen wavefunctions are labeled by an angular momentum: in the 1D representation of C6v point group, there are six elements corresponding to one s-orbital, two p-orbitals, two d-orbitals and one f-orbital, carrying angular momenta $0, \pm \hbar, \pm 2\hbar, \text{ and } 3\hbar$ (with mod 6). All these angular momenta are defined *locally* on the hexagonal unit cell, as can be read explicitly for the two p- and d-orbitals in Fig. 2d, which indicates that they are to be defined as OAM. The two p-orbitals and two d-orbitals form doubly degenerate Dirac dispersions at Γ point of the Brillouin zone (BZ), and reverse their frequencies when the structure is tuned away from the neat honeycomb structure in two opposite ways, which achieves the nontrivial topology. In simulations and experiments, as shown in Figs. 2d and 3, we have checked successfully OAM by measuring the phase winding of EM wavefunctions surrounding the hexagonal unit cell: the EM mode that increases/decreases $\pm 2\pi$ is the p_{\pm} orbital, whereas that increases/decreases $\pm 4\pi$ is the d_{\pm} orbital, respectively.

Moreover, as shown in Fig. 4 in the updated manuscript, where new experimental results are additionally included in the updated manuscript along with the simulation results, OAM is clearly resolved by the circular Poynting vector $\mathbf{r} \times \mathbf{S}$, where the Poynting vector \mathbf{S} is defined *locally* as

$$\mathbf{S} = \frac{|E_z|^2}{2\mu\omega} \left(\frac{\partial\varphi}{\partial x} \hat{\mathbf{x}} + \frac{\partial\varphi}{\partial y} \hat{\mathbf{y}} \right),$$

in terms of the out-of-plane electric field $E_z = |E_z| \exp(i\varphi)$ in the topological interface channel. To the best of our knowledge, no such a local Poynting vector has ever been resolved in microwaves and photonics systems up to this moment. Therefore, our present work reveals in an unambiguous way a new light-matter interaction with local OAM of EM mode hosted by the honeycomb metamaterial structure, and demonstrates that it gives birth to the valuable topological EM transportation.

In our antenna array as the light source, phase delays are introduced between the four rod antennas, which launches EM wave with phase increasing or decreasing surrounding the hexagonal unit cell clockwise, overlapping with, and thus being able to excite, either (p_-, d_-) or (p_+, d_+) modes with fixed sign of OAM exclusively. In the present circumstance SAM is irrelevant, opposing to the guess of this Reviewer. In the last version of Supplement E: Experimental setup, we mentioned the light source with “circular polarization” in order to contrast with linearly polarization (corresponding to the case that all four rod antennas carry the same phase). Because the term of “circular polarization” may cause unnecessary confusions, we remove it in the updated manuscript.

***Comment 2)** As shown in Fig. 4 and described in Ref 27, the selection of the clockwise/anti-clockwise modes is determined by the spin angular momentum (circular polarisation states). However, the authors state in the introduction that the sign of OAM plays the role of an emergent pseudospin degree of freedom, which can mislead and should be justified. In general, there is a close relation between the SAM and OAM (for example, R. Kerber et al., ACS Photonics vol. 4, 891 (2017)), and how the relation affects a light-matter interaction depends on the specific system considered.*

Reply: We appreciate this comment which is related the above one. The selection of the clockwise and counterclockwise modes is determined by the phase decrease and/or increase in the array of antenna rods, which overlaps with, and thus is able to excite, the eigenmodes (p_-, d_-) and (p_+, d_+) with given sign of OAM exclusively. The irrelevance of SAM in the present design becomes clearer when one sees that the OAM-governed unidirectional edge propagations can be excited by linearly polarized source where no SAM exists.

In order to show theoretically that the sign of OAM plays the role of pseudospin in topological interface EM transportation in the present system, we present a $k \cdot p$ theory

as documented in the Supplementary Information in the updated manuscript. The $k \cdot p$ Hamiltonian for the four orbitals of p_{\pm} and d_{\pm} at the Γ point, a 4x4 matrix, is block diagonalized into two 2x2 matrices referring to p_{+}/d_{+} and p_{-}/d_{-} , as far as terms linear in momentum k are considered which contribute to relevant Berry curvatures. This is because that the p_{+} and d_{+} eigen EM modes, and p_{-} and d_{-} ones, differ from each other by $\pm \hbar$, which can be linked to each other by finite off-diagonal matrix elements proportional to $k_{\pm} = k_x \pm ik_y$. This $k \cdot p$ Hamiltonian takes the same form as the BHZ model for the quantum spin Hall effect (Ref. 12), where the two 2x2 blocks are associated with the electronic spin-up and -down channels. Parallelizing these two Hamiltonians, it is clear that the sign of OAM of eigen EM mode in the present photonic crystal plays the same role as the spin in spin-orbit coupled electronic systems, indicating that the sign of OAM works a pseudospin degree of freedom in the present EM system.

In another word, the pseudospin in our setup is the clockwise and/or counterclockwise flux of EM energy which can be given explicitly by the local Poynting vector \mathbf{S} defined above. In a 2D bulk system, the eigenmodes cannot transport any energy since the Poynting vector circulates around the hexagonal unit cell. However, at the interface between a topological regime and a trivial regime, an energy flow appears as demonstrated in Fig. 4 in the updated manuscript where the Poynting vector is not exactly symmetric. Our pseudospin is similar to that proposed and realized by M. Hafezi et al. (see Refs. 24 & 25 in the main text) where ring resonators accommodate clockwise and counterclockwise light modes, except for that ring resonators enjoy the full rotation symmetry whereas C_{6v} symmetry is sufficient in the present setup hosted by the honeycomb structure. As yet another possible formalism for revealing the OAM as a pseudospin degree of freedom in the present design, one can refer to a paper by Brendel et al., PRB vol. 97, 020102 (2018) (Ref. 48 in the updated manuscript).

This Reviewer is right that there is a close relation between the SAM and OAM, and how the relation affects a light-matter interaction depends on the specific system considered, and nice discussions can be found in previous works (see Refs. 39-41 in the updated manuscript). In the present approach, the 1D representation of the point group of C_{6v} symmetry corresponds to the OAM with the rotation axis perpendicular to the microstrip plane, which is the reason that we adopt naturally OAM to describe the physics underlying the device design. We appreciate this Reviewer for raising this comment, and revise our manuscript to put our work in a broader background.

Comment 3) The expression "fine tunable orbital angular momentum" in the title can mislead because the tuning parameter changes the topological state not the OAM. It also gives an impression that the system is actively tunable for example by optical or electrical stimuli, but in fact the tuning parameter is a fixed design parameter that determines which topological regimes a microstrip structure is.

Reply: There seems to be a misunderstanding here. When we discuss the topological interface mode, we have chosen (i.e. fixed) two values $\tau_{1,\text{topo}}(> \tau_0)$ and $\tau_{1,\text{tri}}(< \tau_0)$ and patchworked the two microstrips; a frequency band gap is determined by these two design parameters, where the topological interface EM modes appear. We then sweep the source frequency of linear polarization within this band gap, and measure quantitatively the spectrum weight in the topological interface mode. As shown in Fig. 3, upon sweeping the source frequency one can tune the interface EM modes to carry mostly OAM of $\pm \hbar$ (p orbital like) or $\pm 2\hbar$, (d orbital like) near the two band edges, or a superposition between them with equal weights at the middle of band gap. This tuning is performed after all geometrical parameters as well as materials parameters are fixed, and thus the tuning is *active*. At the present stage, the frequency has to be changed as the price for obtaining EM waves with different OAM. A possible resolution is to use varactors, with which one can adjust the capacitance C at nodes of honeycomb lattice by a static voltage, and thus compensate the variation of frequency. Therefore, we believe that our present work provides a new facet towards final realization of the active tunability. We thank this Reviewer for raising this question. In order to avoid unnecessary confusions, we remove the term of "tune" in the discussions related to the choice of design parameter τ .

Comment 4) It is interesting to see the direct measurement of topological phase transition, pseudo-spin dependent excitation of edge modes for the honeycomb structures but it is not surprising to see them because the mechanism and related symmetry discussions were already included in the paper by Wu and Hu (ref. 27).

Reply: We thank this Reviewer for evaluating our present work. Let us emphasize the once again the main achievements in the present work, which in our opinion are important: 1) The present topological system is based on a 2D circuit, which is compatible with various lithographically fabricated planar devices. In contrary, the previous work by Wu and Hu (ref. 27) was formulated based on dielectric photonic

crystal of infinitely tall cylinders, which, aiming to emphasize the underlying physics, simplifies theoretical analysis. As the price, the design requires fine-controlled fabrications in experimental implementations. With different light-matter interactions, the present work is by no means just an experimental verification of the previous theoretical proposal. 2) Particularly, OAM in the topological EM modes is detected experimentally for the first time and its tunability is demonstrated successfully, which is crucial for attaching additional degree of freedom to EM waves in forthcoming works.

With the ubiquitousness of microstrip TML in EM devices, the present work is expected as a cornerstone which puts hand-in-hand the fundamental interest and novel functionality of EM topology, and will trigger various research activities in the fields of microwave and photonics engineering. The scheme of light-matter interaction demonstrated in the present planar circuitry can be pushed to micro and nano scales where optomechanical waves are used for processing and transformation of quantum information. Therefore, the present work will leave impact to a rich variety of related fields.

We believe that the concerns of this Reviewer have been addressed appropriately, and hope to have her/his recommendation for publication in Nature Communications.

Reviewer #2 (Remarks to the Author):

Comment: Photonic topological insulators have attracted much attention in the past decade. The unique transport properties of edge modes in topological systems allow for robust propagation, immune to scattering from defects and unconventional trajectories such as corners. This robust propagation is very attractive for applications in nano-photonics and photonic circuits, allowing novel designs that were previously thought of as impossible. True topological protection requires breaking of time reversal symmetry, usually done by applying an external magnetic field. However, the magnetic field constitutes a significant hurdle in implement structures, especially in optical domain. While photonic topological insulators without magnetic fields offer topological protection that is not pure, they are more viable as platforms for applications. Photonic topological insulators of both kinds have been experimentally demonstrated by several groups over the past years.

This manuscript presents an experimental demonstration of a crystalline topological insulator; in which the topological protection is based on crystalline symmetry, without breaking time reversal symmetry. The experiments are performed in the microwave regime using microstrip transmission line structure. There are two main novel claims of the paper: One, the implementation of a photonic topological insulator in a microstrip structure, which is an open two dimensional structure allowing for lithographic manufacturing and thus integration to more complex systems. Two, the direct measurement of the intensity and phase of the propagating edge modes, emphasizing the role of OAM in the structure.

Reply: We are grateful to this Reviewer for reading carefully our manuscript. S/he has caught the main messages of our manuscript and put them into the right, prospective scope of the photonic topological insulator, a very active research field.

Comment: Given the increasing interest in photonic topological insulators, I believe the manuscript is of interest to the community. Specifically, structures that allow for future miniaturization and integration are of interest, and this manuscript provides one such example. The manuscript is clearly written and the results are presented in a convincing manner, both to experts in the field and a general audience. Therefore my conclusion is that the manuscript should be published in Nature communications.

Reply: We appreciate the very good assessment given by this Reviewer on our work

and her/his recommendation for publication.

Comment: *Minor points:*

- In line 36 and 37, the authors mention applications for photonics in general. They mention IoT (Internet of Things. I had to search for the acronym; it was not obvious to me) and self-driving automobiles. These seem to me too broad applications, without any immediate relation to metamaterials or photonic circuits. The authors should elaborate.

Reply: We thank this Reviewer for pointing out this, and remove this part from our manuscript.

Comment: *- In figs 3 and 4 the simulation and experimental results are shown in low resolution. While I could understand the results, I had trouble doing so. I assume the finished manuscript will have higher resolution figures so the reader can clearly observe the phase structure of the EM fields.*

Reply: We apologize for this inconvenience and in the revised manuscript we improve the resolution of figures. Please be kindly reminded that, when the word file as submitted is transformed into a pdf file in the submission system, the figure quality is reduced due to automatically software handling. We sincerely hope that this new version does not suffer the same problem.

We thank this Reviewer for her/his positive assessment on our present work and recommendation for publication in Nature Communications. In order to accommodate the comments from this Reviewer we make revisions to our paper, and believe that the manuscript now is ready for publication.

Reviewer #3 (Remarks to the Author):

Comment The authors studied topological effects in planar circuits and realized different relative weight of *p*- and *d*-orbital like EM modes by sweeping frequency. I have no formal objection to the results presented in the manuscript, but this research is a routine development in this area by considering:

1) The theoretical claim of topological band gap in C_6 -symmetric systems had been presented in the same authors; previous PRL paper, and it has been adopted to many different photonic systems, e.g., refs. 28-30, and even phononic systems [e.g., *Nature Physics* 12, 1124 (2016) & *Physical Review Letters* 119, 255901 (2017) & *Physical Review B* 97, 020102 (2018), etc.]. The planar circuits discussed in this manuscript is just another implementation. It is not timely and will not of interest to others in the community and the wider field.

Reply: We thank this Review for reading carefully our manuscript, and for accepting the results presented in the manuscript. As noticed by this Reviewer, the idea of using C_{6v} crystalline symmetry to generate 2D topological photonic and phononic states are getting certain amount of recent interests as a generic prescription for realizing topologically nontrivial states in systems of bosonic feature without breaking time-reversal symmetry.

However, we respectively disagree with her/his comment that the present work is a routine development in this area, and not timely and will not of interest to others in the community and the wider field. Because the design principle is based on C_{6v} symmetry, the present approach does share some common features with a few previous works. However, this does not degrade the importance of the present work for the following reasons. Taking advantage of the planar and open structure of the microstrip, we have successfully detected the orbital angular momentum of EM modes and clarified in a direct experimental way its importance in governing the unidirectional topological interface propagation, which can hardly be achieved in any other electronic and bosonic systems.

As is well known, transmission line (TL) is a basic building block ubiquitous in all electromagnetic devices, and microstrip is a typical TL consisting of metallic substrate, middle dielectric film and top metallic strip. The present work demonstrates clearly that arranging the metallic strip into a honeycomb pattern merely infuses nontrivial Berry

phase and topology into the EM modes. It is remarkable that one does not have to pay any additional price to get the valuable topological features in TL. The planar circuit structure investigated in the present work can be fabricated with conventional techniques, which can be pushed all the way down to nanoscale where advanced lithographical fabrications are already available. All these merits make the approach adaptive to various existing devices, and leave much room for exploration of new functionalities. It is also worth mentioning that the underlying physics is captured by simple LC circuitry, therefore the recipe is accessible to most engineers, and even by high school students. The results are not only useful directly for microwave and photonic engineering, but also suggestive for other circuitry systems including those used for controlling qubits by microwaves or surface mechanical waves. With potential advantages in miniaturization and integration, we believe the present work will trigger **various** applications towards realizing new functionality and thus has far reaching importance.

***Comment:** 2) The experimental results presented in Fig. 4 had been demonstrated in authors' previous work [ref. 30]. Both of them are microwave observation, and once again this results are expected.*

Reply: In contrast to Ref. 30 based on a photonic crystal of dielectric cylinders, the present one is based on a planar microstrip TL, where we have measured successfully for the first time the phase winding of the topological interface modes, which is crucial for revealing experimentally the pseudospin degree of freedom governing the edge mode, and for tuning the relative weights of dipole and quadrupole components.

Moreover, we can map out experimentally the distribution of local Poynting vectors in terms of the amplitude and phase winding of the out-of-plane electric field

$$\mathbf{S} = \frac{|E_z|^2}{2\mu\omega} \left(\frac{\partial\varphi}{\partial x} \hat{\mathbf{x}} + \frac{\partial\varphi}{\partial y} \hat{\mathbf{y}} \right)$$

with $E_z = |E_z| \exp(i\varphi)$, unavailable in all previous works reported so far. The new experimental result is now shown in Fig. 4 in the updated manuscript.

This Reviewer is right that Figs. 4a-d in the last version of our manuscript are similar to those in Ref. 30, which are commonly taken as a direct proof for topological protection of EM propagation. In order to accommodate the comment from this Reviewer, we move them to Supplementary Information in the updated manuscript.

Comment 3) The authors try to claim that the so-called tunable orbital angular momentum by sweeping frequency. However, I doubt the claim of such phenomena can hardly be achieved in other photonic systems in the Abstract. Based on the same theory, similar phenomena should be found in the C6-symmetric all dielectric photonic crystals.

Reply: This would be true if a theoretical understanding is concerned. However, it is clear only from Fig. 3q based on experimental measurements that the relative weights of the p and d orbitals change smoothly with the frequency, where only a small frequency offset between simulations (no pure theoretical results are available) and experimental observation is found. In contrast, in all systems reported so far no experimental results on phase winding of topological interface modes were available, where tunability of OAM could not be claimed in a realistic sense. This Reviewer is right that putting this claim in abstract may cause confusions. We therefore remove this sentence, hoping that our message is conveyed by discussions in the body of our manuscript.

Comment 4) Why is the tunable orbital angular momentum important? There is no answer from the text.

Reply: We thank this Reviewer for raising this important question. EM waves with OAM attract considerable current interests. It becomes clear that optical fields with OAM are ideal for many important applications such as communications, particle manipulation and high-resolution imaging (see for example C.-W. Qiu and Y.-J. Yang, *Science* 357, 645 (2017)) (Ref. 50 in the updated manuscript). Even with microwave frequency, tunable OAM provides a new degree of freedom, which can be used for controlling on-chip propagations and exploited to develop novel information-rich radar and wireless communication protocols (see for example B. Thide et al., *Phys. Rev. Lett.* Vol. 99, 087701 (2007); F. Tamburini et al., *New J. Phys.* Vol. 14, 033001 (2012)) (Refs. 51 and 52 in the updated manuscript).

Comment 5) Orbital angular momentum is important in the optical frequency, can this planar circuit be extended to optical frequency?

Reply: This topological planar circuit can be extended to the middle infrared regime, which covers the important terahertz band, whereas hard to near infrared and visible lights.

Comment 6) *The paper is hard to read and bad to organize. For example:*

i) in Line5 Page4, it is so confused to have optical concept of OAM in the microwave regime. Should clarify the connection between optics and microwave.

Reply: We apologize for this confusion, and revise “optic OAM” into “OAM of EM mode”.

Comment ii) *why the author claimed 2D structure by described the structure consisting of three layers?*

Reply: Although the present setup consists of three layers, the EM wave propagates in the lateral direction, where the magnetic field is aligned in the xy plane and the E_z electric field is confined between the metallic strip network and substrate, and does not vary in the normal direction within the dielectric film. We have demonstrated that OAM defined in 2D plays the crucial role in inducing nontrivial topology in the present microstrip. With all these features said, we agree with this Reviewer that the present system is not a purely 2D system, and revise the manuscript wherever confusions may be caused.

Comment iii) *there is an offset of 0.03G between numerical and experimental results in Fig. 3. But the explanation is due to the tolerance of the material and structural parameters in the fabrication, it is too simple to convince reader.*

Reply: In our microstrip metamaterials, on-node chip capacitors are loaded shunt to a common ground, the whole microstrip system is fabricated on F4B dielectric film. The chip capacitors can vary from the nominal value. Fabrication tolerances in printing the top metallic strip lines onto the F4B dielectric film cannot be avoided. Moreover, the relative permittivity of the F4B dielectric film used for fabricating the microstrip can also slightly deviate from its nominal value. With all these sources said which contributes the discrepancy between simulations and experimental results, we do not think the offset is too large to be acceptable. As a reference, one can see the paper by George V. Eleftheriades et al., PRL, vol. 92,117403 (2004), where an offset of 0.057G is observed at the working frequency of 1G.

Comment iv) *the sum of $(d+)^2 + (p+)^2$ is not equal to 1 [see Fig. 3q]?*

Reply: We thank this Reviewer for raising this question. In order to perform the integrals in Eq. (1SC) numerically, we discretized the regime including a unit cell as shown in the following panel:

We sum up the weights involving phase φ of E_z field and azimuthal angle θ , with θ and 2θ specifying p and d orbitals respectively, over the colored square meshes where E_z field is finite as shown in the above panel. The summations divided by the area of the regime wired by the dash-dot lines in the above panel yield the weights $|p|^2$ and $|d|^2$, which are shown in Fig. 3q when the frequency is swept in the bulk frequency gap. As inspected by direct counting, the area in numerators A_n (namely the whole area of the colored square meshes) is slightly smaller than that in the denominator A_d (namely the whole area of the wired region) with $(A_n/A_d)^2 \approx 0.83$, which counts for the apparent deviation of $|p|^2 + |d|^2$ from unity as presented in our manuscript. To be quantitative, we show in the following the summation of $|p|^2 + |d|^2$ as a function of the frequency. It is clear that the weight summation takes approximately a constant 0.82 with a standard deviation 0.02, consistent with the squared area ratio of 0.83. We notice that the apparent discrepancy between $|p|^2 + |d|^2$ and unity can be reduced by taking finer square meshes in simulations and experiments.

Comment *v)* the amplitude of E_z is missing in the calculation of the weight of p and d orbital states.

Reply: This is an approximation. As can be seen in the following figure, taking into account the amplitude of E_z in the integration does not change the overall behavior.

Comment *Based on the above reasons, I do not recommend this work for publication of Nature Communications.*

Reply: We appreciate this Reviewer for reading carefully our manuscript and raising comments on our work. We believe that the concerns of this Reviewer have been addressed appropriately, and hope to have her/his recommendation for publication in Nature Communications.

Reviewers' comments:

Reviewer #1 (Remarks to the Author):

After reading the author's response and having another careful look at the manuscript, I could gain deeper understanding of their work and what they are trying to present. Before moving onto technical points, I must confess the paper is still difficult to read because of the incorrect use of words and poor organisation rather than the essence of their work (see 4 below). Although this was also pointed out by Reviewer 3, any revision on this direction has not been made. I think significant number of sentences in the main text should be rephrased to be read easily by readers in related fields.

I agree that the physics in dielectric arrays of C_{6v} symmetry can be applied to other structures with C_{6v} symmetry in different wavelength regimes. The authors showed the common features very nicely with the simple equations (Eq. 1-3) for the microstrip structure and demonstrated the topological edge modes experimentally with detailed analysis of the phase of electric field. As the authors answered, the LC circuit model provides a very nice and easy physical insight and the direct measurement of the phase of pseudo-spin up and down is clearly an advance in this hot field. However, there are claims that are unclear and therefore need to be clarified or justified with proper grounds.

1. Definition of OAM is not still clear in the manuscript. According to the response and the paper (Wu and Hu, PRL 2015), the concept of OAM is formulated from the pseudo-spins of local field in a unit cell and therefore can be $0, \pm h, \pm 2h$ and $\pm 3h$ for C_{6v} symmetric structures. However, no accurate definition of OAM (the size and direction) is not given in any of these papers or references. The lack of the clear definition makes me wonder whether the OAM is a well-defined classical vector or a simple analogy of quantum description of electronic orbitals.

2. As mentioned in the outlook, the OAM is certainly a very interesting and important topic in modern applications. However, the OAM in the cited papers is inherently different from the OAM in the paper. The OAM in this paper is a local quantity that is defined for a unit cell and perpendicular to the wavevectors, whereas the OAM in Ref. 51,52 is globally defined for a twisted light (or a vortex beam with a shape of Laguerre-Gaussian beams for example) which has angular momentum parallel to the wavevector. I am not sure whether it is possible to carry the local OAM for a long distance using the proposed design. Moreover, I am not convinced that this would give richer internal degree of freedom as described in the abstract since the quantum numbers of the OAM in this paper is limited by the crystal symmetry whereas the other is not. Therefore, I think the difference needs to be clearly stated not to mislead the readers. Some sentences have been added on the bottom of page 4, but the difference is not clear (although the "local" OAM is mentioned) and the descriptions for two different OAMs are mixed.

3. I do not agree that the OAM is tuneable. I can understand why the authors describe the system as "active" but I don't think the frequency-dependent modal shape can be described as "active". As I mentioned in my comment, I think the general perception of "active" is that some properties can be changed by electrical or optical stimuli. In addition, to use the expression "tuneable", I think an active element should be included in the system. For example, the use of a varactor can make the system tuneable and active as the authors answered, but that is a new system different from the author's one. In this work, I think we can use the word "tuneable" only in the design step where we find the right parameters for each topological regime. Therefore, I think the use of tunable OAM mislead.

4. Below are expressions that are difficult to understand because of the vague/wrong words or wrong orders of words in the first 5 pages. The manuscript needs to be proofread thoroughly.

- Measuring accurately in microwave experiments both amplitude and phase... -> Measuring both

amplitude and phase .. accurately in microwave experiments.

- In the topological interface EM propagations: grammatically wrong.
- Novel functionalities: too vague.
- Up to this moment -> up to now
- Transporting paths and integrated transmissions: What do they mean?
- Their detailed components: not specific.
- Neat honeycomb pattern -> maybe complete honeycomb lattice?
- Implementing a periodic hexagonal pattern of ... band gap: It is not clear why this opens a bandgap.
- We measure in terms of near-field techniques....-> We measure the distributions of bothusing near-field techniques...
- Along the interface channel -> what is interface channel?
- In contrary, -> In contrast,

I think the work is very interesting and provides a good physical insight to a topological system. However, based on the points in the above, I cannot recommend it to be accepted for publication for Nature Communications without clarification on the first three points and a significant change in the descriptions of the results.

Reviewer #2 (Remarks to the Author):

I have received the corrected manuscript form the authors.

After reading the other reviewers comments and the authors' response, I maintain my previous assessment: the papers' main novelty is in its experimental design, which allows for direct measurement of the EM field in topological structures and cal lead to future miniaturization to NIR frequencies. I believe such novelty is enough for publication in Nature Communications, as the paper in my eyes is clearly written and the results look promising.

Reviewer #3 (Remarks to the Author):

I have read the response from the authors, but I'm sorry that this manuscript does not reach the high-quality of Nature Communications.

Just as presented in my original comment, this work is not timely by considering authors' previous theoretical works in Phys. Rev. Lett. 114, 223901 (2015) and recent experimental work in Phys. Rev. Lett. 120, 217401 (2018).

In the reply, the authors try to claim that 'Taking advantage of the planar and open structure of the microstrip, we have successfully detected the orbital angular momentum of EM modes and clarified in a direct experimental way its importance in governing the unidirectional topological interface propagation, which can hardly be achieved in any other electronic and bosonic systems'. But I'm sure similar phenomena can be achieved in the structure used in Phys. Rev. Lett. 120, 217401 (2018). By changing the input sources, unidirectional topological interface propagation can definitely be achieved in any other bosonic systems.

In the reply, the authors also try to claim that 'The planar circuit structure investigated in the present work can be fabricated with conventional techniques, which can be pushed all the way down to nanoscale where advanced lithographical fabrications are already available'. This is an experimental work at the microwave range. The results presented in Figs. 3 and 4 are expected, but not exciting or attractive. If these results are extended to the middle infrared regime by fabricating the nanoscale samples (just like what the author reply), then the results are good and fancy because the nanofabrication and measurement are not easy jobs. But for a microwave results (by considering authors' similar work in PRL2018), I'm sorry the novelty is not strong.

(Please be reminded that all the page numbers and reference numbers appear in this document are those of the updated version of the manuscript, including the main text and supplementary information.)

Reviewer #1 (Remarks to the Author)

***Comment:** After reading the author's response and having another careful look at the manuscript, I could gain deeper understanding of their work and what they are trying to present. Before moving onto technical points, I must confess the paper is still difficult to read because of the incorrect use of words and poor organisation rather than the essence of their work (see 4 below). Although this was also pointed out by Reviewer 3, any revision on this direction has not been made. I think significant number of sentences in the main text should be rephrased to be read easily by readers in related fields.*

Reply: We thank this Reviewer for careful reading of our manuscript and response, and her/his insightful comments on our work. We are happy to improve our manuscript accordingly, which should make our work more accessible to readers in related fields.

***Comment:** I agree that the physics in dielectric arrays of C_{6v} symmetry can be applied to other structures with C_{6v} symmetry in different wavelength regimes. The authors showed the common features very nicely with the simple equations (Eq. 1-3) for the microstrip structure and demonstrated the topological edge modes experimentally with detailed analysis of the phase of electric field. As the authors answered, the LC circuit model provides a very nice and easy physical insight and the direct measurement of the phase of pseudo-spin up and down is clearly an advance in this hot field. However, there are claims that are unclear and therefore need to be clarified or justified with proper grounds.*

Reply: We sincerely thank this Reviewer for her/his positive assessment on our work. As for unclear claims and/or definitions pointed out by this Reviewer, we wish to clarify one-by-one in what follows, and revise our manuscript to accommodate the comments.

For details please see the following replies and the revised manuscript, as well as the summary of changes.

Comment: 1. Definition of OAM is not still clear in the manuscript. According to the response and the paper (Wu and Hu, PRL 2015), the concept of OAM is formulated from the pseudo-spins of local field in a unit cell and therefore can be $0, \pm h, \pm 2h$ and $\pm 3h$ for C_{6v} symmetric structures. However, no accurate definition of OAM (the size and direction) is not given in any of these papers or references. The lack of the clear definition makes me wonder whether the OAM is a well-defined classical vector or a simple analogy of quantum description of electronic orbitals.

Reply: First of all, we thank this Reviewer for raising this comment on OAM, which is the crucial point of the present approach. A clear definition of OAM can be given, which originates from the LC lumped element circuitry resembling to the tight-binding description of electrons, and is measured as a related classical vector in the present microstrip setup, as revealed in what follows.

Let us first explain the OAM in the present crystalline circuitry of C_{6v} symmetry. As can be found in standard text books for group theory, OAM is related to the 1D irreducible representation for the point group of C_{6v} . In the present LC lumped element model with C_{6v} symmetry, the eigen wave functions at Γ point are given by $\{\exp[i l \theta_j]\}$ where θ_j is the azimuthal angle with $j = 1 \sim 6$ referring to the sites in the hexagonal unit cell, and l is the quantum number of OAM (see Fig. 2 of the manuscript). These are equivalent to the wave functions in the tight-binding model of electrons as given in Ref. 47, bearing in mind the equivalence between the present LC lumped element circuitry and the TB model (see Eqs. (1~3) in the manuscript), and that the wave functions of an isolated hexagon are the same as those at Γ point for a triangle lattice of hexagons with inter-hexagon hoppings. The quantum number l takes $0, \pm 1, \pm 2$ and 3 in the present system of C_{6v} symmetry with modulus 6, and the wave functions associated with $l = \pm 1$ and ± 2 correspond to p_{\pm} and d_{\pm} .

In microstrips investigated in the present work, the EM field distributes over the strips

continuously, with the symmetry features of the EM modes and their impacts to the nontrivial topology remaining unchanged from the LC lumped element model. As the evidence, the dependences of the phase of out-of-plane electric field on the *continuous* azimuthal angle for two typical eigen wave functions are shown in the following figure, which are the full-wave-simulation results for the microstrip with parameters same as those for the topological microstrip in Fig. 3 in the main text. It is clear that, on the lower/upper frequency band edge, the eigen wave function is given by $E_z = |E_z| \exp(i\varphi) = |E_z| \exp(il\theta)$ with $l = 2$ and $l = 1$ respectively, where θ is the continuous azimuthal angle. This is in complete agreement with the picture given by the LC lumped element circuitry.

For continuous distributions of EM fields, OAM can be defined as a classical quantity in terms of the circulating local Poynting vector given in Eq. (4) in the manuscript. In the present microstrip it reads

$$\mathbf{L} = \mathbf{r} \times \mathbf{S}/c^2 = \frac{|E_z|^2}{2\mu\omega c^2} \left(x \frac{\partial \varphi}{\partial y} - y \frac{\partial \varphi}{\partial x} \right) \hat{\mathbf{z}},$$

where the position vector \mathbf{r} is measured from the center of unit cell. For the eigen wave functions given in the above figure, one has

$$\mathbf{L} = \frac{|E_z|^2}{2\mu\omega c^2} l \hat{\mathbf{z}}.$$

Dividing this OAM by the energy density [see L. Allen, M. W. Beijersbergen, R. J. C. Spreeuw and J. P. Woerdman, “Orbital angular momentum of light and the transformation of Laguerre-Gaussian laser modes”, Phys. Rev. A vol. 45, 8185 (1992)], we arrive at

$$\frac{\mathbf{L}}{\varepsilon|E_z|^2/2} = \frac{l\hbar\hat{\mathbf{z}}}{\hbar\omega},$$

namely one photon carries a quantized OAM $l\hbar$ in the direction perpendicular to the microstrip plane. Therefore, the OAM defined by the eigen wave functions based on the LC lumped element model, resembling to those in electronic systems, and that defined by the circulating local Poynting vector, being able to be measured experimentally, are equivalent to each other.

The above discussions together with the results presented in the manuscript clarify the crucial role of OAM in realizing the topological states by C_{6v} crystalline symmetry, and that the scenario applies for EM states, such as those in the microstrip structure under consideration, where the physical quantities are continuous. While the picture was given in the previous paper Ref. 27, the present microstrip structure provides a platform where the circulating local Poynting vectors can be measured experimentally with high accuracy, which bridges the two definitions of OAM in an explicit way.

It is noticed that at the interface, where C_{6v} symmetry is broken partially, OAM is not a good quantum number anymore. This corresponds to the continuous variation of the weights of p_{\pm} and d_{\pm} orbitals observed in our experiments when the frequency is swept through the frequency band gap. It is clear, however, that the main features of the OAM are inherited from the bulk to the topological interface modes as a manifestation of bulk-edge correspondence in topological states, and, inversely, the measured OAM carried by the topological interface modes reveals the bulk properties in a unique way.

In order to accommodate the valuable comment from this Reviewer, we revise the manuscript. For details please refer to the new version of manuscript, both main text and supplementary information (Supplements D and F), and the summary of changes.

Comment: 2. As mentioned in the outlook, the OAM is certainly a very interesting and important topic in modern applications. However, the OAM in the cited papers is inherently different from the OAM in the paper. The OAM in this paper is a local quantity that is defined for a unit cell and perpendicular to the wavevectors, whereas the OAM in Ref. 52, 53 is globally defined for a twisted light (or a vortex beam with a

shape of Laguerre-Gaussian beams for example) which has angular momentum parallel to the wavevector. I am not sure whether it is possible to carry the local OAM for a long distance using the proposed design. Moreover, I am not convinced that this would give richer internal degree of freedom as described in the abstract since the quantum numbers of the OAM in this paper is limited by the crystal symmetry whereas the other is not. Therefore, I think the difference needs to be clearly stated not to mislead the readers. Some sentences have been added on the bottom of page 4, but the difference is not clear (although the “local” OAM is mentioned) and the descriptions for two different OAMs are mixed.

Reply: This Reviewer is right that the OAM explored in the present work is perpendicular to the propagating direction of the topological interface EM modes, and thus is apparently different from those in the literature which are parallel to the propagating direction. While the OAM in the present work is defined on unit cell, and is local in this sense, it can propagate in the photonic crystal for long distance as far as losses in metallic stripes can be neglected, which is a good approximation for microwaves, and therefore can carry information inside the system formed by the crystalline microstrip.

How to emit efficiently an EM wave carrying OAM supported by the microstrip structure with C_{6v} symmetry into free space is one of the most intriguing questions with the properties revealed by this work in hand. We would like to leave it for future investigations, which is expected to pave the way for designing topological emitters of microwaves with OAM as the intrinsic degree of freedom.

Following the comment from this Reviewer, we revise the descriptions on the OAM in our system and those in literature and make their differences clear. For details please see the new version of manuscript and the summary of changes.

Comment: 3. *I do not agree that the OAM is tuneable. I can understand why the authors describe the system as “active” but I don’t think the frequency-dependent modal shape can be described as “active”. As I mentioned in my comment, I think the general*

perception of “active” is that some properties can be changed by electrical or optical stimuli. In addition, to use the expression “tuneable”, I think an active element should be included in the system. For example, the use of a varactor can make the system tuneable and active as the authors answered, but that is a new system different from the authors’ one. In this work, I think we can use the word “tuneable” only in the design step where we find the right parameters for each topological regime. Therefore, I think the use of tunable OAM mislead.

Reply: We agree with this Reviewer on this point and revise our discussions in the manuscript accordingly. Especially, we change the title of our manuscript into “Topological LC circuitry based on planar microstrip and observation of orbital angular momentum in interface electromagnetic modes”. For details please see the new version of manuscript and the summary of changes.

Comment: 4. Below are expressions that are difficult to understand because of the vague/wrong words or wrong orders of words in the first 5 pages. The manuscript needs to be proofread thoroughly.

- *Measuring accurately in microwave experiments both amplitude and phase;*
- > *Measuring both amplitude and phase .. accurately in microwave experiments.*
- *In the topological interface EM propagations: grammatically wrong.*
- *Novel functionalities: too vague.*
- *Up to this moment -> up to now*
- *Transporting paths and integrated transmissions: What do they mean?*
- *Their detailed components: not specific.*
- *Neat honeycomb pattern -> maybe complete honeycomb lattice?*
- *Implementing a periodic hexagonal pattern of band gap: It is not clear why this opens a bandgap.*
- *We measure in terms of near-field techniques -> We measure the distributions of bothusing near-field techniques;*
- *Along the interface channel -> what is interface channel?*
- *In contrary, -> In contrast,*

Reply: We thank this Reviewer for raising the detailed suggestions. We revise our manuscript accordingly. For the details please refer to our updated manuscript and the summary of changes.

Comment: I think the work is very interesting and provides a good physical insight to a topological system. However, based on the points in the above, I cannot recommend it to be accepted for publication for Nature Communications without clarification on the first three points and a significant change in the descriptions of the results.

Reply: We appreciate very much this Reviewer for her/his positive assessment on our work, and the suggestions on how to improve the manuscript. Especially, we have revised our manuscript in the way to address the first three points raised in her/his report. The comments of this Reviewer really help us a lot in sharpening the presentation of our results, and revisions based on the comments make our work much more accessible by readers in the related fields. We sincerely hope to have her/his recommendation for publication of our work in Nature Communications.

Reviewer #2 (Remarks to the Author)

Comment: I have received the corrected manuscript form the authors. After reading the other reviewers comments and the authors' response, I maintain my previous assessment: the papers' main novelty is in its experimental design, which allows for direct measurement of the EM field in topological structures and can lead to future miniaturization to NIR frequencies. I believe such novelty is enough for publication in Nature Communications, as the paper in my eyes is clearly written and the results look promising.

Reply: We really thank this Reviewer for her/his persistent positive assessment on our work. The comments on our original design of experimental setup and its potential for future miniaturization to NIR frequencies are very encouraging. Her/his

recommendation for publication of our work in Nature Communications is highly appreciated.

Reviewer #3 (Remarks to the Author)

Comment: I have read the response from the authors, but I'm sorry that this manuscript does not reach the high-quality of Nature Communications.

Reply: We thank this Reviewer for reading our manuscript and response. As for her/his comments on the quality of our work, we wish to explain one by one in what follows.

Comment: Just as presented in my original comment, this work is not timely by considering authors' previous theoretical works in Phys. Rev. Lett. 114, 223901 (2015) and recent experimental work in Phys. Rev. Lett. 120, 217401 (2018). In the reply, the authors try to claim that "Taking advantage of the planar and open structure of the microstrip, we have successfully detected the orbital angular momentum of EM modes and clarified in a direct experimental way its importance in governing the unidirectional topological interface propagation, which can hardly be achieved in any other electronic and bosonic systems". But I'm sure similar phenomena can be achieved in the structure used in Phys. Rev. Lett. 120, 217401 (2018). By changing the input sources, unidirectional topological interface propagation can definitely be achieved in any other bosonic systems.

Reply: We respectfully disagree with this Reviewer on this point. In order to satisfy the preconditions of the theory given in Ref. 27 for achieving topological photonic states based on dielectric cylinders, one either makes cylinders very tall, or puts the cylinders into two metallic plates perpendicular to the cylinders with a reasonable separation. In Ref. 31 the second approach is adopted, where Al_2O_3 cylinders are mounted to the bottom metallic plate, and the top one is put slightly above the cylinders, such that it can be slid during measurement. The electric field was measured which evidences the trajectory of topological EM modes along the interface between the topological and trivial photonic crystals. However, because the phase of electric field is influenced

significantly by the metallic plate, one could not map out the local Poynting vector accurately (which is given by a formula similar to Eq. (4) where the phase of electric field is involved), in contrast to the microstrip setup investigated in the present work. Likewise, while unidirectional propagations of topological interface modes can be detected in various bosonic systems based on measurements of field strengths, it is not always possible to map out the energy flow accurately. Our present setup based on microstrip provides the first example where the detailed pseudospin states, pseudospin-momentum locking and the p-d hybridization in the topological interface EM propagation can be demonstrated precisely.

The novelty of the present work is captured by both Reviewer #1 and #2: Reviewer #1 put it clear in her/his second report that the LC lumped element circuitry provides a very nice and easy physical insight, and that the direct measurement of the phase of pseudo-spin up and down is clearly an advance in this hot field. Reviewer #2 mentioned, in consistency with her/his first report, that the main novelty of our work is in its experimental design, which allows for direct measurements of the EM field in topological structures and can lead to future miniaturization to NIR frequencies.

***Comment:** In the reply, the authors also try to claim that “The planar circuit structure investigated in the present work can be fabricated with conventional techniques, which can be pushed all the way down to nanoscale where advanced lithographical fabrications are already available”. This is an experimental work at the microwave range. The results presented in Figs. 3 and 4 are expected, but not exciting or attractive. If these results are extended to the middle infrared regime by fabricating the nanoscale samples (just like what the author reply), then the results are good and fancy because the nanofabrication and measurement are not easy jobs. But for a microwave results (by considering authors’ similar work in PRL2018), I’m sorry the novelty is not strong.*

Reply: The present work provides a design principle for achieving topological EM states in versatile transmission lines, which can be described by a simple LC circuitry, and smoothly connected to various electronic systems. We demonstrate successfully that the theoretical proposal is realized perfectly in experiments by showing the precise

measurement results on distributions of both strength and phase of the electric field, and the energy flow given by local Poynting vectors, with the latter never been achieved in any topological systems. Our present work will trigger future works in various transmission lines and other circuitry systems, including those involving nanoscale fabrications. Therefore, the present work has sufficient novelty, and potential impacts in related fields.

We have addressed all the comments from this Reviewer, and believe that the work is now ready for publication in Nature Communications.

REVIEWERS' COMMENTS:

Reviewer #1 (Remarks to the Author):

The authors clarified the ambiguous points and improved the manuscript in both presentation of results and English expressions. I am happy with the changes the authors have made. Therefore, I recommend the manuscript to be accepted for publication in Nature Communications.